# Review of Piezoelectrical Materials Potentially Useful for Peripheral Nerve Repair

**DOI:** 10.3390/biomedicines11123195

**Published:** 2023-12-01

**Authors:** Diogo Casal, Maria Helena Casimiro, Luís M. Ferreira, João Paulo Leal, Gabriela Rodrigues, Raquel Lopes, Diogo Lino Moura, Luís Gonçalves, João B. Lago, Diogo Pais, Pedro M. P. Santos

**Affiliations:** 1Departamento de Anatomia, NOVA Medical School|Faculdade de Ciências Médicas (NMS|FCM), Universidade NOVA de Lisboa, 1169-056 Lisboa, Portugal; luis.soldado.g@gmail.com (L.G.); diogo.pais@nms.unl.pt (D.P.); 2Plastic and Reconstructive Surgery Department and Burn Unit, Centro Hospitalar Universitário de Lisboa Central, Rua José António Serrano, 1169-045 Lisbon, Portugal; 3Centro de Ciências e Tecnologias Nucleares (C2TN), Instituto Superior Técnico (IST), Universidade de Lisboa, 2695-066 Bobadela, Portugal; casimiro@ctn.tecnico.ulisboa.pt (M.H.C.); psantos@ctn.tecnico.ulisboa.pt (P.M.P.S.); 4Departamento de Engenharia e Ciências Nucleares (DECN), Instituto Superior Técnico (IST), Universidade de Lisboa, 2695-066 Bobadela, Portugal; ferreira@ctn.tecnico.ulisboa.pt; 5Centro de Química Estrutural (CQE), Institute of Molecular Sciences (IMS), Instituto Superior Técnico (IST), Universidade de Lisboa, 2695-066 Bobadela, Portugal; jpleal@ctn.tecnico.ulisboa.pt; 6Centro de Ecologia, Evolução e Alterações Ambientais (cE3c) & CHANGE—Global Change and Sustainability Institute, Departamento de Biologia Animal, Faculdade de Ciências, Universidade de Lisboa (FCUL), 1749-016 Lisboa, Portugal; mgrodrigues@ciencias.ulisboa.pt; 7Gynaecology and Obstetrics Department, Maternidade Alfredo da Costa, Centro Hospitalar Universitário de Lisboa Central, R. Viriato 1, 2890-495 Lisboa, Portugal; raquelmgslopes@gmail.com; 8Anatomy Institute and Orthopedics Department, Faculty of Medicine, University of Coimbra, 3004-504 Coimbra, Portugal; dflmoura@gmail.com; 9Spine Unit, Orthopedics Department, Coimbra University Hospital, 3000-602 Coimbra, Portugal; 10Departamento de Biologia Animal, Faculdade de Ciências, Universidade de Lisboa (FCUL), 1749-016 Lisboa, Portugal; joaolago133@gmail.com

**Keywords:** peripheral nerve, repair, electroactive scaffolds, electrospinning, piezoelectric polymers, piezostimulation, biodegradables, biomedical devices, surgery, 3D printing

## Abstract

It has increasingly been recognized that electrical currents play a pivotal role in cell migration and tissue repair, in a process named “galvanotaxis”. In this review, we summarize the current evidence supporting the potential benefits of electric stimulation (ES) in the physiology of peripheral nerve repair (PNR). Moreover, we discuss the potential of piezoelectric materials in this context. The use of these materials has deserved great attention, as the movement of the body or of the external environment can be used to power internally the electrical properties of devices used for providing ES or acting as sensory receptors in artificial skin (e-skin). The fact that organic materials sustain spontaneous degradation inside the body means their piezoelectric effect is limited in duration. In the case of PNR, this is not necessarily problematic, as ES is only required during the regeneration period. Arguably, piezoelectric materials have the potential to revolutionize PNR with new biomedical devices that range from scaffolds and nerve-guiding conduits to sensory or efferent components of e-skin. However, much remains to be learned regarding piezoelectric materials, their use in manufacturing of biomedical devices, and their sterilization process, to fine-tune their safe, effective, and predictable in vivo application.

## 1. Introduction

The piezoelectrical effect was discovered in 1880 by the French scientists Jacques and Pierre Curie. They observed that piezoelectrical materials can transfer electrons when pressed and/or twisted, and receive back these electrons when distended or relaxed, allowing the generation of electric dipoles. The observation that pressure on certain materials (single-crystal quartz in the original description) generated an electrical charge led to the term “piezoelectricity” being coined (“piezo” means “pressure” in Greek). Piezoelectricity results from the conversion of mechanical energy (mechanical strain and vibration) into electric polarization without the need of applying an external voltage. Since its original description, its use has become extensively widespread, and there now exist numerous applications of this effect in industrial, military, domestic and health care settings [1,2,3]. 

The qualification of the piezoelectric effect can be made through the Piezoelectric Coefficient, which can be defined as the charge (expressed in Coulombs) developed on the surface of the piezoelectric material per unit force applied on it (expressed in Newtons). Hence, in the SI system, the unit becomes coulomb/Newton. Notwithstanding, since the charge developed per unit of force is small, the Piezoelectric Coefficient is more conveniently expressed as pC/N [4]. 

Interestingly, more recently it was noted that piezoelectricity was ubiquitous in nature and particularly in living beings. It can be observed in the various degrees of nature’s hierarchical organization, from the amino acid and protein levels to DNA molecules, viruses, tissues, organs, skeletons, and even jungles and seashores [5,6,7,8,9]. 

Moreover, it has increasingly been realized that electrical currents may play a pivotal key role in cell migration and tissue repair. Hence, the heightened interest in piezoelectrical materials, particularly in the realm of peripheral nerve repair (**PNR**), where the contemporary reconstructive strategies present frequently dismaying results, leaving those affected with permanent motor, sensory and/or autonomic disability. Furthermore, these patients are often stricken with neuropathic pain whose treatment is often difficult and incomplete [10,11,12,13]. Therefore, peripheral nerve injuries (**PNI**) exert a significant psychosocial and economic burden on both individuals and society [13,14].

This is all the more important taking into consideration that PNI are relatively common from birth to old age, occurring in multiple contexts, namely compressive neuropathies, multiple types of trauma, as a result of tumor treatment, in anesthetic procedures, infections or degenerative diseases [14,15,16,17,18,19]. Up to one in every 1000 children are born with a significant PNI (brachial plexus palsy) [20]. It is estimated that in Sweden alone, each year there are 13.9 new cases for each 100,000 people of serious PNI mandating hospitalization [14]. In the USA, the mean annual incidence of PNI is 16.9 per 100,000 for the upper extremity, and 13.3 cases per million for the lower extremity [10]. In this country, more than 200,000 trauma-related nerve injuries occur each year [21]. Worldwide, the incidence of PNI in the head, neck and trunk regions is also significant, although difficult to quantify [22]. 

In this review, we will summarize the current evidence supporting the potential benefits of electric stimulation (**ES**) in the physiology of PNR. Subsequently, we will discuss the materials with piezoelectrical properties available for producing devices with potential use for PNR. Next, we will review the devices already described using piezoelectrical properties for PNR. Finally, we will discuss future perspectives concerning the use of piezoelectrical materials in this context.

## 2. Role of Electrical Stimulation in the Physiology of Peripheral Nerve Repair

It has been known for more than 150 years, since the seminal works of Luigi Galvani and later on of those of Emil Du-Bois Reymond, that after tissue injury there is a local disturbance in electric charges, generating endogenous electrical fields and electric current [23,24,25]. These, in turn, create electric dipoles that guide and promote the migration of numerous cell types in process named “galvanotaxis” or “electrotaxis” [24,26,27,28]. Numerous animal studies have shown that galvanotaxis is initiated immediately after injury, helping coordinate all the processes (hemostasis, inflammation, proliferation and remodeling) that lead to definitive tissue repair [24,26,27,28]. Under the influence of electric fields, peripheral nervous system neurons extend protrusions and migrate towards the cathodic pole [26,29]. This type of behavior has also been observed in human fibroblasts, lymphocytes, macrophages [30], and endothelial cells, all of which are known to be important in peripheral nerve repair [30,31,32,33]. Piezoelectric materials can generate electrical charges in response to mechanical strain, thus stimulating axonal regeneration by galvanotaxis following nerve injury [34,35]. To increase the amount of electricity produced by piezoelectric materials, some authors have applied external ultrasound sources to internally placed devices that are used in PNR [36,37,38].

In 1952, Hoffman noted enhanced peripheral nerve regeneration after applying ES stimulation in nerve roots [13,39]. Subsequent studies on rabbit and rat hindlimb models confirmed the regenerative-inducing potential of ES in the peripheral nervous system [40,41,42,43].

The reason why the application of ES either intra-operatively or post-operatively has not yet become broadly accepted is probably because its mechanisms of action have remained largely elusive until recently. In fact, only in the past years have several technological advances allowed accurate electrophysiological measurements close to the injury site and contributed to the unravelling of the underlying physiological mechanisms behind enhanced PNS recovery after ES [13,23,26,44,45].

However, even today multiple questions remain unanswered. The most primordial and pressing question probably pertains to the mechanisms that allow cells to sense electrical charges [23]. Notwithstanding, several studies have suggested that asymmetrically distributed cell receptors, namely, integrins and receptors of acetylcholine, epidermal growth factor and of concanavalin A, probably play a role in the electrotactic response [23,34,35,46,47,48,49,50,51,52,53].

Experimental data suggest that ES is transduced by the second messenger molecules cyclic AMP, Rho-associated protein kinase and phosphoinositide-3 kinase [29]. Additionally, ES causes up-regulation of brain-derived neurotrophic factor, T alpha-1 tubulin, growth-associated protein 43 (GAP-43), as well as other regeneration-associated genes, resulting in axon regeneration [35,45,54,55,56]. Globally, all these events lead to increased neuronal cell adhesion, proliferation, migration, and protein synthesis, particularly of neuronal cytoskeletal proteins, hastening the outgrowth of PNS axons across the injury site [11,12,23,35,43,45,54,57,58]. Additionally, ES promotes remyelination of elongating axons by Schwan cells [45,59]. Furthermore, in a mouse model, ES has been shown to induce differentiation of neural stem cells and progenitor cells into neurons and glial cells [26,60].

Clinically, ES has been studied sparsely for the past decades. Most studies are related to its percutaneous application for prevention of muscular atrophy after PNS injury [61]. There have been four randomized clinical trials on the use of ES in the clinical setting, all presenting positive results. Two of them report postoperative ES after carpal tunnel and cubital tunnel surgical release [62,63]. Another study describes the prevention of accessory nerve dysfunction after oncologic neck dissection using intraoperative ES [64]. Lastly and most revealingly, there is paper on the brief post-surgical low frequency ES of surgically repaired digital nerves which had been accidentally sectioned. This work showed accelerated axon outgrowth across the repair site and hastened target reinnervation [54].

All these data spurred the recent enthusiasm over the use of electrical currents to treat different types of pathologies, including PNS lesions. This, in turn, led to the term “electroceuticals” being coined [26,34,35]. Lack of sound data regarding the treatment of PNI with ES, namely concerning the best method of delivery of ES, its frequency, duration or intensity, the occasional discomfort associated with its use in awake patients, the need for multiple interventions, and its feasibility in critical nerve gap injury have generally been enumerated as reasons for the lack of general acceptance of this method [12,23,26,32,34,35].

In a significant number of PNI cases, there is a gap between the nerve stumps that precludes their surgical suture. In these cases, it is necessary to apply a conduit to “bridge” the nerve defect. In fact, after a nerve section there is a latency period of up to 30 days in which the proximal axons do not elongate in the direction of the distal nerve stump [32]. If the nerve gap was not protected and bridged, the surrounding connective tissue would proliferate in this period and physically block most of the elongating axons from the proximal stump from reaching the distal nerve stump. Traditionally, autologous dispensable nerves or veins are used to bridge nerve defects [65,66,67]. More recently, allogenic nerve grafts and artificial nerve-guiding conduits (**NGC**) were introduced, and are in widespread use [13,32]. However, none of these options is perfect. Autologous alternatives entail non-negligible donor site morbidity. Allogenic nerve grafts have been associated with inflammatory reactions akin to rejection responses. Generally, both allogenic nerve grafts and NGC have been associated with worse functional results than autologous nerve grafts or flaps, particularly for longer nerve defects [13,16,32,35,65,66,67,68].

Hence, great effort has been put into developing better NGCs, ideally with ES properties. In this context, the use of piezoelectric materials has deserved great attention, as the movement of the body could be used to power internally the electrical properties of the device, avoiding toxic batteries which eventually need to be removed or exchanged surgically [69,70,71,72,73,74].

The authors propose that there is enough evidence to believe that piezoelectric materials may play a significant role in the treatment of PNI. In this paper, the authors will try to provide a critical appraisal of the literature on this subject.

## 3. Piezoelectrical Materials

There is a plethora of both inorganic and organic strongly piezoelectric materials available to construct NGCs and other PNI repair devices [5,74]. When choosing these materials, it is fundamental to have a sound grasp of not only their piezoelectric properties, but also their safety profile, their biocompatibility, their biostability, and their degradation products inside the body. Moreover, it is also important to understand the techniques available to shape them into the desired geometrical configurations. Finally, their ductility, resistance and softness should also be controlled, in order for the devices to be fixed with sutures and tolerated inside the body [35,74] (Table 1). 

## 4. Inorganic

Aluminum nitride (AlN), Barium titanate (BaTiO3), Lead zirconate titanate (PZT-5H) are examples of biocompatible ceramic materials with a high piezoelectric response. However, they are rigid, brittle, and contain non-degradable and toxic compounds that limit their potential for constructing implantable devices [5,99].

One of the most commonly used inorganic polymers in PNR is polyvinylidene fluoride (PVDF). This compound is flexible, has excellent piezoelectric properties and is biocompatible, allowing for direct contact with biological tissues. However, it is not degradable, requiring a removal surgery with all the inconvenience and the risks it entails [5,74].

Graphene is another compound that, despite having piezoelectric properties, a high surface area and high electrical conductivity, can be hazardous when inserted into the body, as it breaks up and its fragments can accumulate inside various organs, potentially causing severe cellular damage and disease [5,74,100].

Metals are considered too rigid to be used alone in ES devices. However, metal nanoparticles, namely of gold, silver, and copper, can be used to increase the mechanical strength and electrical conductivity of composite materials. Some metals can progressively dissolve, such as magnesium, zinc, tungsten, iron and molybdenum, allowing the construction of “transient electronics” [101]. However, due to the consumption of oxygen and release of byproducts in the corrosion of these metals, which can lead to adjacent tissue necrosis, biosafety studies are warranted before clinical trials are implemented [5,35,74,102,103,104,105].

## 5. Organic

Organic piezoelectric biomaterials can be of different classes, such as natural occurring amino acids (e.g., glycine, cysteine, alanine, threonine, diphenylalanine), proteins (e.g., collagen, silk), and polysaccharides (e.g., cellulose, chitin, chitosan, alginate), or synthetic polymeric compounds, such as poly-lactic acid (PLA), Glycine-Polyvinyl alcohol (PVA), Polycaprolactone (PCL), Polyamide (PA), and Polypyrrole (PPy) [106,107,108]. Since synthetic polymers have greater mechanical qualities than natural polymers and can be readily synthesized into 3D structures, they are frequently employed to fabricate NGCs for PNR [109,110].

Although with a lower piezoelectric effect compared to many inorganic materials, these organic compounds present a much more favorable biocompatibility, biosafety, and biodegradability profile. In fact, these organic compounds are readily recognized and naturally degraded by host cells and/or microbiome enzymes, allowing for recipient cell invasion and progressive replacement of the device with endogenous tissues. Hence, living tissues can easily tolerate these compounds without triggering unfavorable immunological reactions. PA and PCL, for example, are frequently used when strength, flexibility and durability are required [111]. Arguably, these properties make these materials the most obvious candidates for the construction of implantable ES devices [69,70,71,110,112].

β-Glycine has received great attention, due to its high piezoelectric constant in its crystalline form. Unfortunately, glycine salts are readily soluble in body fluids and are difficult to handle, as they are hard and brittle. To circumvent these limitations, β-Glycine crystals have been associated with Polycaprolactone to form a soft and resistant material with a significant piezoelectric effect. This composite material has already been used with success as devices placed inside rodents’ brains, and to produce NGC [5,74,113].

Collagen is the most common extracellular protein in animals. It makes for a good material for PNR as it promotes cell adhesion and development and presents excellent biocompatibility, hydrophilicity, and low antigenicity. Moreover, it has significant piezoelectric properties [78,114,115].

Silk fibroin is extracted from silkworm silk and has exceptional mechanical strength, flexibility, and biocompatibility. These characteristics enable the production of silk fibroin in a variety of shapes, including films, fibers, and scaffolds, all of which can be adjusted to closely resemble the mechanical characteristics of peripheral nerves. Silk fibroin is a perfect substance for nerve tissue engineering, since it has also been demonstrated to support cell adhesion, proliferation, and differentiation [79].

Cellulose is a biocompatible and biodegradable polymer, with piezoelectric properties, that has been increasingly used in PNR. Cellulose has exceptional mechanical qualities, such as high strength and stiffness, which are essential for supporting the structure of the nerve during regeneration. This is crucial for PNR, since the material needs to be able to endure mechanical stresses and maintain its integrity until axonal elongation is concluded. Furthermore, cellulose has good biocompatibility. This quality is necessary to support cell adhesion, differentiation, and proliferation—all processes that are necessary for effective neuron regeneration. Moreover, cellulose is a sustainable and renewable substance, which makes it an attractive choice from an environmental standpoint. Its abundance in nature and ability to be derived from various sources, such as plants or bacteria, further contribute to its appeal as a piezoelectric material for peripheral nerve repair [80]. 

Chitin can be obtained from the exoskeletons of insects, arthropods, and crustacean shells. After cellulose, chitin is the most common natural polysaccharide. Chitosan can be found in some fungi or be derived from chitin through the partial deacetylation of the latter [79,116].

Chitin and chitosan possess several advantageous features, namely biocompatibility, biodegradability, amenability to create various geometrical forms (porous scaffolds, hydrogels, fibers, sponges, films, etc.), chemical and enzymatic modifiability, antimicrobial characteristics, potential for controlled release of cytokines, antibiotics and extracellular matrix constituents, the ability to promote cell adherence and viability [82]. The characteristics mentioned above led to multiple studies on the potential use of chitosan and chitin in reconstruction of peripheral nerves [117].

Another interesting natural polysaccharide is alginate. This compound is present in the cell walls of brown algae. It is characterized by its hydrophilicity, retaining large volumes of water and forming gels in the process. Alginate is also characterized by its biocompatibility and low toxicity. Its main disadvantages are uncontrollable and relatively fast degradation, inadequate mechanical strength, and inadequate cell signaling [118]. Furthermore, alginate piezoelectric properties are minimal [83].

Polyvinylidene fluoride (PVDF) is a frequently utilized synthetic organic piezoelectric material for PNR, due to its piezoelectric qualities. It possesses adequate mechanical flexibility and strength, electrical conductivity, and biocompatibility for the production of NGC. However, this material is naturally strongly hydrophobic, which limits its usage in biomedical device fabrication, unless chemically modified or associated with other more hydrophilic materials [119,120].

Alternatively, poly(lactic-co-glycolic acid) (PLGA), is a biodegradable and biocompatible polymer, which possesses piezoelectric properties when aligned in a certain direction. Its biodegradability has been exploited for the controlled release of drugs or growth factors, further enhancing the regenerative process [121]. 

Additional synthetic organic piezoelectric materials used in the realm of PNR are polyurethane (PU) and poly(3-hydroxybutyrate-co-3-hydroxyhexanoate) (PHBHHx). PU exhibits good mechanical properties, biocompatibility, and electrical conductivity. PHBHHx, on the other hand, is a biodegradable and biocompatible polymer that can be electrospun into nanofiber scaffolds for nerve regeneration [86,87,122].

Poly-*γ*-benzyl-L-glutamate (PBLG) is another synthetic peptide. It is a resilient and flexible material with good mechanical qualities, simplicity of processing, and flexibility, which make it a good fit for applications involving PNR. In animal models, PBLG showed enhanced axonal outgrowth and nerve regeneration. Furthermore, scaffolds based on PBLG have been created, offering structural support to direct nerve cells and promote their proliferation. By imitating the natural extracellular matrix, these scaffolds can be made to encourage cell adhesion, proliferation, and differentiation [5].

The fact that most organic materials sustain spontaneous degradation inside the body means their piezoelectric effect is also limited in duration. In the case of PNR, this is not necessarily problematic, as ES is only required during the regeneration period of the PNS [69,70,71,72,74,102].

However, in general, it can be argued that hardly any given material possesses ideal biological, piezoelectric and structural properties. For instance, naturally occurring materials with piezoelectric qualities present better biocompatibility and bioactivity. Conversely, synthetic piezoelectric materials such as PVDF and lead zirconate titanate (PZT) can be engineered to have specific mechanical and piezoelectric features. However, synthetic materials tend to be long-lasting, often behaving as a foreign body that may entrap the growing nerve and/or increase the risk of infection. These in turn may mandate a second surgery for scaffold extraction, increasing cost and potential morbidity [70]. Finally, graphene and other highly piezoelectric materials have been added to other more readily rebsorbable materials to enhance the piezoelectric properties of the composite materials [123].

Thus, combining different natural compounds, natural and synthetic materials, or different synthetic compounds in composite materials may allow the fine-tuning of the composition of devices for optimal PNR. For example, a recent study showed improved neurite outgrowth and nerve regeneration with a composite scaffold NGC made of PVDF and collagen. While the natural collagen aided in cell attachment and tissue integration, the synthetic PVDF supplied the required mechanical support and enhanced piezoelectric qualities. Similar studies have been performed with a myriad of natural and/or synthetic compounds [70,85,124,125,126]. 

## 6. Processing Piezoelectrical Materials

The piezoelectric effect of materials results from the regular alignment of molecular dipoles [127,128]. In some cases, it may be useful to increment this effect, particularly when the processing of the material diminishes the original piezoelectric effect. There are several methods to obtain this increment, namely stretching of the material (drawing), thermal annealing (heat treatment), application of a high external electrical field (electrical poling) or maximizing the macroscopic alignment of fibers in the case of materials obtained through electrospinning processes [5,74].

Stretching materials, especially at high temperatures, multiple times, promotes the alignment of dipoles. Heat treatment increases the crystalline content of amorphous materials, increasing their piezoelectric properties. Electrical poling consists of applying a high-voltage electrical field (commonly 1 to 10 kV) to ferroelectric materials (i.e., materials that have spontaneous electric polarization that can be reversed by the application of an external electric field), in order to align dipoles, which, in turn, will increase their piezoelectric properties and ensure the desired polarization [5,68,129]. 

Some authors have augmented the piezoelectric effect of devices by externally applying ultrasound to drive this effect [130]. Notwithstanding, detractors of this process argue that the frequent need for acoustic streaming may have deleterious effects in tissues, namely through cavitation and local heat generation [35,36,131].

## 7. Biomedical Devices

Several biomedical devices with piezoelectric properties have been applied experimentally for PNR [53]. 

Piezoelectric materials can be used to produce tridimensional scaffolds amenable to the implantation of cells important in the process of PNR, such as elongating axons, Schwan cells or mesenchymal cells. Bio-printed devices may even include the latter two cell types in their composition. Additionally, substances required for chemically stimulating cell growth and differentiation may be added. Finally, some of these chemotactic substances may also have piezoelectric properties, such as chitosan [5,68,116,132,133,134,135,136]. Organic bioresorbable piezoelectric materials are good options for tissue engineering, due to their biocompatibility, minimal toxicity and galvanotaxis effect [5,132].

Nerve-guiding conduits (NGC) are the most reported devices. They are used to bridge nerve gaps, which occur relatively frequently in clinical practice. Traditionally, these gaps have been reconstructed with resort to autologous nerve grafts, which entails variable donor site morbidity and limited supply [137]. 

Their potential association with specific cell therapy, growth factors, gene therapy alone or in combination, has been placing NGC as a promising alternative to nerve autografts [35,71,138,139,140]. NGC can be produced by various methods, namely 3D printing, mold casting, electrospinning, and roll-up sheeting [35,112]. Three-dimensional printing of NGC allows design freedom, as well as the possibility to replicate complex nerve anatomy in monolithic devices without any assembly requirements [112,141]. 

A self-powered patch composed of a flexible piezoelectric generator applied over a wound bed has been shown to promote skin nerve regeneration and sensation [142].

Electronic skin, also known as e-skin, is a broad term used to refer to artificial skin that emulates human skin, not only for covering and protective purposes, but also for providing haptic, thermal and humidity sensations [5,143,144]. It has a wide range of potential applications, namely robotics, prosthetics, virtual reality, human/machine interfacing, monitoring vital signs, detecting environmental pollutants, and human skin replacement [144,145].

E-skin typically consists of three main components: a flexible substrate, functional materials, and sensors. The flexible substrate provides the base for the electronic circuitry and ensures its mechanical flexibility. Various materials, such as polymers or nanomaterials, are used to achieve the desired properties of flexibility, stretchability, and durability. In several conceptions of e-skin, receptors are piezoelectric, although capacitive and resistive receptors have also been described [144,145].

Despite its multiple potential advantages, e-skin also presents several challenges. One limitation is the difficulty in achieving long-term stability and reliability of the electronic components embedded within the skin-like material. The mechanical and electrical properties of the e-skin need to be carefully optimized to ensure durability and performance over time. Additionally, the scalability and manufacturing processes of e-skin need to be further developed to enable mass production at a low cost. Moreover, the integration of power sources, such as batteries or energy harvesters, remains a challenge for e-skin devices [146].

## 8. Discussion

Even today, despite countless surgical and technological advancements, the clinical results after PNR remain unsatisfactory. This is surprising and certainly unrelated to the immense time scientists have devoted to research in this field. In fact, visionary surgeons like Paul of Aegina apparently were performing nerve sutures as far back as 600 AD [67,147,148,149,150,151].

Hence, the use of piezoelectric materials in the realm of PNR holds great promise as a breakthrough technology that may improve clinical results. In fact, by allowing the conversion of mechanical energy from normal movements of the body into electrical gradients, or the translation of the contact with normal external stimuli into electrical potentials, these materials can be used to produce scaffolds, NGC or e-skin. The ES they produce can be channeled to drive PNR. In fact, these materials have been shown in some experimental and clinical studies to promote the growth and alignment of nerve fibers that are regenerating. Moreover, many of these materials, particularly organic ones, are biocompatible, lowering the possibility of rejection or inflammation and ensuring compatibility with biological tissues [34,35].

Additionally, several of the devices produced with piezoelectric materials can be 3D-printed [112,152,153]. This manufacturing technique presents an enormous versatility and room for creativity. Being based on CAD (Computer Aided Design) files and not having the normal constraints of traditional manufacturing methods, it allows a wide range of geometric forms, including highly complex organic structures. Additionally, CAD files can be further refined using generative design. This process can be defined as a set of computational methods, including artificial intelligence algorithms and machine learning, designed to maximize structural performance requirements with the minimal amount of material, and a faster printing speed. Generative design can be used to perform topological optimization, reinforcing structures in the regions where greater forces are applied, without the need to create continuous objects or surfaces. In medical devices, this lattice structure reduces weight and facilitates native tissue invasion and integration. Simultaneously, this architecture promotes survival of cells by simple diffusion initially and by neo angiogenesis subsequently. Therefore, 3D printing has the potential to greatly improve the ergonomics and efficiency of medical devices used in PNR. Moreover, 3D-printing biomedical devices allows design freedom and the possibility to replicate complex nerve anatomy in monolithic devices without any assembly requirements. For example, printing NGCs before surgeries could diminish resort to autologous nerve grafts and their associated morbidity [137,154,155,156]. 

The inner structure of NGC can be printed in a compartmentalized fashion, providing additional physical clues to guide elongating axons [157,158].

Finally, 3D-printed piezoelectrical devices may even be associated with growth factors and/or cells, further boosting PNR [112,152,159,160,161,162,163].

However, there are still significant hurdles and difficulties related to the restoration of peripheral nerves using piezoelectric materials. Fabricating scaffolds or conduits with the mechanical strength, pliability, biocompatibility, nontoxicity, durability, and piezoelectric characteristics (ES dosage and polarity) required for implantation in the human body is one of the key issues. Advanced production processes and exact material composition and structure control are needed for this [34,35,85].

A potential caveat of using piezoelectric materials to produce NGC is the requirement of an adequate orientation of the electric polarization of the device. In fact, as mentioned above, neurons, fibroblasts, macrophages, and endothelial cells, paramount in PNR, have been shown experimentally to migrate towards the cathodic pole [26,29,30,31,32,33,164]. Electrical poling and stretching of materials can be used to align dipoles [5,68,129]. Characterization of piezoelectric features with Piezoresponse Force Microscopy and/or Atomic Force Microscopy measurements will help establish the efficacy of these techniques in specific devices [68,145,165,166]. 

Additionally, a largely overlooked aspect in this field is the effect of the required sterilizing processes required prior to in vivo implantation of biomedical devices. One of the most common ways to sterilize medical devices is using gamma radiation. However, its effects on the structure, physical and biological properties of commonly used 3D-printed devices is largely unknown [167]. There is some evidence that gamma radiation may cause weakening of mechanical properties, namely of tensile strength and elongation, which could limit the practical use of the 3D-printed devices. This knowledge is therefore of paramount importance to uphold the requirements for medical devices’ safety and usefulness [168,169,170]. 

Furthermore, it has been shown that PNR is dependent on local blood supply [66,67]. Hence, long and/or wide NGCs may provide inadequate blood supply to the elongating nerve. Therefore, increasing devices’ porosity or producing prefabricated vascularized NGCs have been proposed to try to mitigate these limitations [66,67,171,172]. In general, these modifications follow the trend of trying to replicate an optimized neuronal microenvironment, including structural, biochemical, electrical, vascular, and biological clues that support and promote PNR [173].

In the future, devices must be thoroughly studied not only in hindlimb models, but also in forelimb models where PNI are more common, and for which data are not yet available [65,67]. Finally, further studies are warranted to confirm or dismiss the promising experimental data and the scarce clinical data on the use of piezoelectric materials for PNR [35,53,65,67].

## 9. Conclusions

Arguably, piezoelectric materials have the potential to revolutionize the somewhat stalled field of peripheral nerve repair with new biomedical devices that range from scaffolds and NGC to sensory or efferent components of artificial skin (e-skin).

However, much remains to be learned regarding the piezoelectric materials, the manufacturing of the biomedical devices, and their sterilization process to fine-tune its safe, effective and predictable in vivo application [35].

## Figures and Tables

**Table 1 biomedicines-11-03195-t001:** Summary of features of different piezoelectric materials used in peripheral nerve repair.

Type			Materials	Biocompatibility	Biodegradability	Mechanical Properties	Pyezolectric Properties	References
**Inorganic**								
			Aluminum Nitride (AlN),	+	+	++, rigid, brittle	+	[75]
			Barium titanate (BaTiO3)	+	+	+++, hard, fracture resistant	+++	[76]
			Lead zirconate titanate (PZT-5H)	+	++	+++, hard, fracture resistant	+++	[77]
			Polyvinylidene fluoride (PVDF)	++	+	++, flexible	++	[74]
			Graphene (G)					
**Organic**								
	**Natural**							
		**Amino acids**	Glycine	+++	+++	+, readily soluble in the body; hard and brittle	++	[5]
			Cysteine	+++	+++	+, readily soluble in the body; hard and brittle	+	[5]
			Alanine	+++	+++	+, readily soluble in the body; hard and brittle	+	[5]
			Threonine	+++	+++	+, readily soluble in the body; hard and brittle	+	[5]
			Diphenylalanine	+++	+++	+, readily soluble in the body; hard and brittle	+	[5]
		**Proteins**	Collagen	+++	+++	++	+	[78]
			Silk	+++	++	+++, exceptional mechanical strength and flexibility	+	[79]
		**Polysaccharides**	Cellulose	+++	++	+++, excellent strength and flexibility	+	[80]
			Chitin	+++	++	+++, high strength and stiffness	+	[81]
			Chitosan	+++	++	++, pliable	++	[82]
			Alginate	+++	++	+, fragile	+	[83]
	**Synthetic**		Poly-lactic acid (PLA),	+++	++	++, rigid and brittle	+	[84]
			Polyvinyl alcohol (PVA)	+++	+++	+++, soft	+	[85]
			Polycaprolactone (PCL)	+++	+++	+++, soft	+	[86]
			Polyamide (PA)	+++	++	+++, flexible, resistant	+	[85]
			Polypyrrole (PPy)	+	+	+++, pyroelectric properties	+	[86]
			Polyurethane (PU)	+	+	++, flexible, resistant	+	[85]
			Poly(3-hydroxybutyrate-co-3-hydroxyhexanoate) (PHBHHx)	++	++	++, flexible, resistant	++	[87]
			Poly-*γ*-benzyl-L-glutamate (PBLG)	+++	+++	+++, flexible, resistant	+++	[88]
**Composites ^1^**								
	**Natural**		Collagen/Tyramine Hyaluronic Acid derivative (HA-Tyr) Hydrogel	+++	+++	+++	+++	[89]
			Silk fibroin/Alginate (SF/Alg)	+++	+++	+++	+++	[90]
			Chitosan/Silk fibroin	+++	+++	+++	+++	[91]
			Chitosan/Collagen	+++	+++	+++	+++	[92]
	**Natural/Synthetic**		Collagen/PCL	+++	+++	+++	+++	[93]
			Chitosan/PCL	+++	+++	+++	+++	[94]
	**Synthetic**		G/PCL	+++	+++	+++	+++	[95]
			G/PPy/PLA	+++	+++	+++	+++	[96]
			PVDF/PCL	+++	+++	+++	+++	[97]
			PVDF/G	+++	+++	+++	+++	[98]

^1^ Composite materials have tunable compositions which allow a better match of the biological and physical properties of the devices. +, poor; ++, good; +++, excellent.

## Data Availability

All the data described are present in the article.

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
