# Peer review of "(untitled)"

_biomedicines, 2023, doi:10.3390/biomedicines11123195_

Round 1
Reviewer 1 Report
Comments and Suggestions for Authors
The authors reviewed piezoelectrical materials potentially useful for peripheral nerve repair. Overall, this review is very lengthy and relatively comprehensive. Here are some comments for consideration.
1. It is long review article. The authors are suggested to display a table to summarize the categories of different piezoelectric materials for PNI.
2. What is the potential effecting mechanisms of piezoelectric materials in the process of nerve regeneration? It should be discussed as this is a vital part of this review.
3. Some important work in the field of piezoelectric materials and fiber materials for peripheral nerve regeneration should be discussed in this paper, e.g. Nano Energy 2021; 83: 105779; Small 2020; 16(32): e2000796; Adv Fiber Mater. 2022; 4:203–213; Adv. Fiber Mater., 2023, 5, 349; Adv. Fiber Mater., 2022, 4, 503; Also, the importance of restoring microenvironment in peripheral nerve regeneration should be emphasized as well (e.g., Mater Today 2021; 51: 165-187).
4. Please discuss the future of the development and clinical translation of piezoelectric materials in the treatment of peripheral nerve injury.
Author Response
“It is long review article. The authors are suggested to display a table to summarize the categories of different piezoelectric materials for PNI.”
In order to comply with this request, the authors added on page 4, a new table (Table 1) summarizing and comparing the different materials used for peripheral nerve repair.
- What is the potential effecting mechanisms of piezoelectric materials in the process of nerve regeneration? It should be discussed as this is a vital part of this review.
The authors added on page 3:
“Piezoelectric materials can generate of electrical charges in response to mechanical strain, thus stimulating axonal regeneration following nerve injury.(35, 36) To increase the amount of electricity produced by piezoelectric materials, some authors have applied external ultrasound sources to internally placed devices that are used in PNR.(37-39)”
Further on, the authors wrote (page 3, lines 117-133):
“However, even today multiple questions remain unanswered. The most primordial and pressing question probably pertains to the mechanisms that allow cells to sense electrical charges.(23) Notwithstanding, several studies have suggested that asymmetrically distributed cell receptors, namely, integrins and receptors of acetylcholine, epidermal growth factor and of concanavalin A probably play a role in the electrotactic response.(23, 46-51)
Experimental data suggest that ES [electrical stimulation] is transduced by the second messenger molecules cyclic AMP, Rho-associated protein kinase and phosphoinositide-3 kinase.(29) Additionally, ES causes up-regulation of brain-derived neurotrophic factor, Talpha-1 tubulin, growth associated protein 43 (GAP-43), as well as other regeneration-associated genes, resulting in axon regeneration.(36, 45, 52-54) Globally, all these events lead to increased neuronal cell adhesion, proliferation, migration, and protein synthesis, particularly of neuronal cytoskeletal proteins, hastening the outgrowth of PNS axons across the injury site.(10, 11, 23, 36, 45, 52, 55-57) Additionally, ES promotes remyelination of elongating axons by Schwan cells.(45, 58) Furthermore, in a mouse model, ES has been shown to induce differentiation of neural stem cells and progenitor cells into neurons and glial cells.(26, 59)”
- Some important work in the field of piezoelectric materials and fiber materials for peripheral nerve regeneration should be discussed in this paper, e.g. Nano Energy 2021; 83: 105779; Small 2020; 16(32): e2000796; Adv Fiber Mater. 2022; 4:203–213; Adv. Fiber Mater., 2022, 4, 503; Also, the importance of restoring microenvironment in peripheral nerve regeneration should be emphasized as well (e.g., Mater Today 2021; 51: 165-187).
The authors have incorporated all the mentioned citations in pertinent sections of the manuscript.
Concerning the role of the microenvironment in peripheral nerve regeneration, the authors added (page 9, lines 441 to 444):
“In general, these modifications follow the trend of trying to replicate an optimized neuronal microenvironment, including structural, biochemical, electrical, vascular, and biological clues that support and promote PNR [peripheral nerve repair].(160)”
- Please discuss the future of the development and clinical translation of piezoelectric materials in the treatment of peripheral nerve injury.
In order to comply with this request, the authors wrote at the end of the Discussion Section (page 9, lines 421 to 449):
“A potential caveat of using piezoelectric materials to produce NGC is the requirement of an adequate orientation of the electric polarization of the device. In fact, as mentioned above, neurons, fibroblasts, macrophages, and endothelial cells, paramount in PNR, have been shown experimentally to migrate towards the cathodic pole.(26, 29-31) (32-34) Electrical poling and stretching of materials can be used to align dipoles. (4, 114, 115) Characterization of piezoelectric features with Piezoresponse Force Microscopy and/or Atomic Force Microscopy measurements will help establish the efficacy of these techniques in specific devices.(115, 133, 149, 150)
Additionally, a largely overlooked aspect in this field is the effect of the required sterilizing processes required prior to in vivo implantation of biomedical devices. One of the most common ways to sterilize medical devices is using gamma radiation. However, its effects on the structure, physical and biological properties of commonly used 3D printed devices is largely unknown.(151) There is some evidence that gamma radiation may cause weakening of mechanical properties, namely of tensile strength and elongation, which could limit the practical use of the 3D printed devices. This knowledge is therefore of paramount importance to uphold the requirements for medical devices safety and usefulness.(152-154)
Furthermore, it has been shown that PNR is dependent on local blood supply.(65, 66) Hence, long and/or wide NGCs may provide inadequate blood supply to the elongating nerve. Therefore, increasing devices porosity or producing prefabricated vascularized NGCs have been proposed to try to mitigate these limitations.(65, 66, 155, 156) In general, these modifications follow a trend of trying to replicate an optimized neuronal microenvironment, including structural, biochemical, electrical, vascular, and biological clues that support and promote PNR.(160)
In the future, devices must be thoroughly studied not only in hindlimb models, but also in forelimb models where PNI are more common, and for which data is not yet available.(64, 66) Finally, further studies are warranted to confirm or dismiss the promising experimental data and the scarce clinical data on the use of piezoelectric materials for PNR.(36, 64, 66, 118)”
Reviewer 2 Report
Comments and Suggestions for Authors
Dear colleagues!
Thanks for the interesting post, however I miss the scientific analysis in your work. In addition to listing the properties of materials, it is necessary to take a critical look at their characteristics.
It would be appropriate if you give a null hypothesis for your study.
In discussion mode, I think it would be correct to make tables where you should compare groups of drugs, their disadvantages and advantages. In addition to textual representation, systematization is also needed.
You also need to rethink the list of references and remove or update outdated sources older than 15 years
Author Response
Thanks for the interesting post, however I miss the scientific analysis in your work. In addition to listing the properties of materials, it is necessary to take a critical look at their characteristics.
We strove to take this criticism into the consideration when revising the manuscript. As mentioned above, we added on page 4 Table 1, which describes and compares different piezoelectric materials used in peripheral nerve repair.
In addition, the authors added (page 5, lines 209 to 218)
“Since synthetic polymers have greater mechanical qualities than natural polymers and can be readily synthesized into 3D structures, they are frequently employed to fabricate NGCs for PNR.(85, 86)
Although with a lower piezoelectric effect compared to many inorganic materials, these organic compounds present a much more favorable biocompatibility, biosafety, and biodegradability profile. In fact, these organic compounds are readily recognized and naturally degraded by host cells and/or microbiome enzymes allowing for recipient cell invasion and progressive replacement of the device with endogenous tissues. Hence, living tissues can easily tolerate these compounds without triggering unfavorable immunological reactions.”
Finally, the authors have added information on each piezoelectric material throughout the manuscript (highlighted areas).
“It would be appropriate if you give a null hypothesis for your study.”
In order to add this information, the authors wrote on page 4, lines 170-172:
“The authors propose that there is enough evidence to believe that piezoelectric materials may play a significant role in the treatment of PNI. In this paper the authors will try to provide a critical appraisal of the literature on this subject.”
Moreover, in the Conclusions Section, the authors wrote (page 9, lines 451-456):
“Arguably, piezoelectric materials have the potential to revolutionize the somewhat stalled field of peripheral nerve repair with new biomedical devices that range from scaffolds and NGC to sensory or efferent components of artificial skin (e-skin).
However, much remains to be learned regarding the piezoelectric materials, the manufacturing of the biomedical devices, and their sterilization process to fine tune its safe, effective and predictable in vivo application.(36)”
“In discussion mode, I think it would be correct to make tables where you should compare groups of drugs, their disadvantages and advantages. In addition to textual representation, systematization is also needed.”
In order to comply with this request, the authors added on page 4, a new table (Table 1) summarizing and constrasting the different materials used for peripheral nerve repair.
“You also need to rethink the list of references and remove or update outdated sources older than 15 years.”
We thoroughly reviewed the references, as indicated, replacing most of the sources older than 15 years for more recent ones. We only left a few older references that report original descriptions of concepts and/or innovations of historical interest.